# OpenReview forum: "Tool-Planner: Task Planning with Clusters across Multiple Tools"
_ICLR.cc/2025/Conference — ICLR 2025 Poster_

### Official Review · Reviewer_uydA · 2024-10-25

**Soundness:** 3
**Presentation:** 3
**Contribution:** 3
**Rating:** 6
**Confidence:** 4

**Summary:**

This paper introduces Tool-Planner, a framework for task planning that uses clustered toolkits as nodes in tree search.

**Strengths:**

1. The idea of using toolkits as nodes in tree-search is novel and intuitive, providing a more structured approach to handling API failures.
2. The experimental results are comprehensive, covering multiple datasets and comparing against various strong baselines.

**Weaknesses:**

1. The method's performance heavily depends on the quality of the clustering algorithm and model. While Table 5 shows end-to-end performance comparisons of different clustering models, there's a lack of intermediate metrics to evaluate clustering quality.

2. The selection of the optimal k value for k-means clustering remains challenging. As is shown in Figure 3, determining an appropriate k value requires exploration and this is actually impractical when facing new datasets.

3. The fundamental contribution of Tool-Planner can be viewed as introducing a prior into conventional tree-search-based methods - specifically, the assumption that when an API fails, the model should first attempt similar APIs. While this intuition is reasonable and the experimental results are promising, the paper has two key limitations in this aspect:

    **a.** Lacking theoretical analysis or justification for why clustering tools into toolkits leads to better performance. While empirically effective, understanding the theoretical underpinnings could provide insights into when and why this approach works better than alternatives.

    **b.** The paper doesn't explore whether similar benefits could be achieved through simpler means. Notably, the same prior (trying similar APIs first when one fails) could potentially be incorporated directly into LLM prompting. It remains unclear whether the additional complexity of explicit clustering and toolkit-based search is necessary, or if similar performance gains could be achieved by simply instructing the LLM to prefer similar APIs when handling failures.

**Questions:**

See weaknesses. Further:

1. Have the authors considered using more sophisticated clustering methods that don't require pre-specifying the number of clusters?
2. How does Tool-Planner perform when the available APIs change frequently? Is reclustering needed, and if so, how often?

I would consider revising my rating based on the authors' responses to these questions.

---

> ### Author Response · Authors · 2024-11-23
> **Response to Reviewer uydA (1/3)**
>
> Thank you to the reviewers for evaluating our paper and providing many important and critical suggestions, as well as some praise. The compliments included recognition of our novel ideas, intuitive presentation, a more structured approach, comprehensive experimental results, and comparisons with various strong baselines, among other aspects. Next, I will provide detailed responses to the concerns you have raised.
>
> > __W1__: The method's performance heavily depends on the quality of the clustering algorithm and model. While Table 5 shows end-to-end performance comparisons of different clustering models, there's a lack of intermediate metrics to evaluate clustering quality.
>
> Thank you for raising this important question. In our approach, since tools with the same functionality need to be grouped into the same cluster to form toolkits, the quality of clustering is critical. The experiments in Table 5 aim to demonstrate that the SimCSE method we adopt can better compress the documentation of tools, essentially comparing the performance of different encoders. Regarding the intermediate metrics for evaluating clustering quality that you mentioned, we have elaborated on them in detail in Table 3.
>
> For example, in scenarios with a larger number of clusters, our method requires a longer reasoning chain, which results in error accumulation during the reasoning process and subsequently leads to a decrease in the win rate. On the other hand, when the number of clusters is smaller, tools within the same cluster may serve different purposes. Consequently, this clustering cannot effectively handle the original planning scheme, leading to a significant decline in quality. The experiments in Table 3 clearly highlight the importance of clustering quality.
>
> Additionally, we evaluated the performance of clustering algorithms such as DBSCAN, and the results are as follows. we use cosine similarity as the distance metric for DBSCAN. Default minpts is set to 1.
>
> Win Rate:
>
> |                            | $$\epsilon$$ | G1-Inst  | G2-Inst  | G3-Inst  | TorchHub | HuggingFace | TensorFlow |
> |----------------------------|----------|----------|----------|----------|----------|-------------|------------|
> | Tool-Planner(k-means best) | -        | 63.3     | **72.3** | 67.5     | 77.6     | 75.4        | **72.4**   |
> | Tool-Planner(DBSCAN)       | 0.001    | 57.5     | 68.3     | 66.3     | 75.1     | 72.8        | 65.2       |
> | Tool-Planner(DBSCAN)       | 0.01     | 61.8     | 71.8     | 66.5     | 77.3     | 76.5        | 70.3       |
> | Tool-Planner(DBSCAN)       | 0.02     | **64.3** | 70.3     | **68.0** | **78.5** | **77.2**    | 71.9       |
> | Tool-Planner(DBSCAN)       | 0.04     | 63.5     | 69.8     | 68.3     | 77.0     | 73.1        | 69.7       |
> | Tool-Planner(DBSCAN)       | 0.08     | 59.5     | 59.3     | 66.0     | 73.2     | 65.8        | 61.2       |
> | Tool-Planner(DBSCAN)       | 0.16     | 50.5     | 45.5     | 57.5     | 65.1     | 56.9        | 55.9       |
>
> It can be observed that under stricter constraint conditions, the model maintains relatively strong performance. However, excessively high constraint conditions reduce the number of tools that meet these constraints, causing the method to degrade, similar to the DFSDT effect. Under appropriate clustering constraints, the tools within a cluster perform the same function. This means that when encountering a subtask, all tools in the cluster can handle the corresponding task. As a result, without frequently altering the reasoning process of the large language model, the model can exhibit improved performance.
>
> > __W2__: The selection of the optimal k value for k-means clustering remains challenging. As is shown in Figure 3, determining an appropriate k value requires exploration and this is actually impractical when facing new datasets.
>
> Selecting the optimal $k$ value for K-means clustering is indeed challenging. However, the determined $k$ value ensures that the relative latency and performance during the reasoning process can be adjusted and managed to some extent. Methods like DBSCAN, which rely on the similarity of data to determine whether points belong to the same cluster, may exhibit uncontrollable performance on new data when the dataset has significant biases.
>
> To some extent, the corresponding differences in clustering information tend to be unstable. On datasets with relatively balanced distributions and tasks, the ideal number of clusters is approximately 1/10 to 1/20 of the total number of tools. Moreover, in multi-API scenarios, our clustering method demonstrates its advantages more effectively. Experimental results using methods like DBSCAN also empirically prove the effectiveness of clustering. Determining the optimal clustering often involves a peak function. By using a binary search approach, the optimal value can be identified more efficiently.

---

> ### Author Response · Authors · 2024-11-23
> **Response to Reviewer uydA (2/3)**
>
> > __W3-a__: Lacking theoretical analysis or justification for why clustering tools into toolkits leads to better performance. While empirically effective, understanding the theoretical underpinnings could provide insights into when and why this approach works better than alternatives.
>
>
> Regarding your question on why clustering tools into toolkits leads to better performance, we have addressed this in the introduction and abstract of the original paper. First, overly complex reasoning processes can lead to error accumulation, resulting in unstable reasoning and execution times.
>
> From a more straightforward perspective, when a system like ToolLLM’s DFSDT approach selects the optimal tool from a pool of over 10,000 tools, this process often involves multiple API call failures and inherent hallucination issues. These issues necessitate changes to the pre-defined reasoning strategy. For large language models, this typically means adding new nodes to the existing plan or breaking down some of the original nodes into multiple steps for resolution. Consequently, solving a major problem results in an average number of reasoning steps significantly greater than a solution with a relatively stable reasoning path.
>
> For Tool-Planner, ensuring such a stable path involves grouping APIs with similar functionalities. When an API fails, the system can rely on previously available information to quickly generate the documentation for the next API and continue reasoning based on the initial plan, rather than switching to a completely new plan. This approach dramatically reduces accumulated errors and, compared to other methods, enhances overall reasoning performance through a more stable reasoning strategy. Therefore, clustering tools is both an effective and crucial method for improving performance.
>
> > __W3-b__:  The paper doesn't explore whether similar benefits could be achieved through simpler means. Notably, the same prior (trying similar APIs first when one fails) could potentially be incorporated directly into LLM prompting. It remains unclear whether the additional complexity of explicit clustering and toolkit-based search is necessary, or if similar performance gains could be achieved by simply instructing the LLM to prefer similar APIs when handling failures.
>
>
> Thank you very much for your suggestion, such as incorporating failed APIs into the LM's prompt and attempting similar APIs based on the same prior knowledge. This type of approach is indeed used in previous methods like Reflection and solutions such as AnyTool, which also adopt feedback-based adjustment strategies. However, these methods fail to address a core issue: although feedback is integrated, there is still a significant likelihood that the overall planning path will be altered by the large language model during the feedback process.
>
> Once such changes occur, the number of sub-tasks in the overall reasoning process increases, and the reasoning path becomes longer. These compounded errors are likely to propagate to the final result. In contrast, clustering-based methods can more quickly aggregate tools with similar functions, providing multiple alternatives for performing a specific function, thereby achieving relative stability in the overall planning process.
>
> Nonetheless, we have compared our method with the DFSDT+Reflective framework, incorporating your suggestion to prompt the model to use similar APIs whenever possible. We conducted a comprehensive evaluation of various metrics, as shown in the table below.
>
> |                          | G1-Inst | G2-Inst | G3-Inst | TorchHub | HuggingFace | TensorFlow |
> |--------------------------|---------|---------|---------|----------|-------------|------------|
> | DFSDT                    | 58.3    | 70.5    | 68.0    | 72.0     | 60.7        | 69.6       |
> | DFSDT(Reflective on ICL) | 59.0    | 68.8    | 67.5    | 73.5     | 61.3        | 68.2       |
> | Tool-Planner             | $63.3$    | $72.3$    | 67.5    | $77.6$     | $75.4$        | $72.4$       |
>
> Experimental results demonstrate that this preprocessing directly improves the overall tool reasoning performance and the final reasoning capability.

---

> ### Author Response · Authors · 2024-11-23
> **Response to Reviewer uydA (3/3)**
>
> > __Q1__. Have the authors considered using more sophisticated clustering methods that don't require pre-specifying the number of clusters?
>
> Yes, we have provided experimental results for DBSCAN, as shown above in Q1. This type of clustering method primarily relies on the similarity between tools themselves and does not require pre-specifying the number of clusters. However, specifying the number of clusters offers the advantage of controlling the overall inference speed, ensuring good performance within a certain range.
>
> > __Q2__. How does Tool-Planner perform when the available APIs change frequently? Is reclustering needed, and if so, how often?
>
> When available APIs change frequently, if we already have information about all previously available APIs, clustering does not need to be adjusted in such cases. This is because the situation essentially involves a reduction in the number of available APIs under a given functionality, while the overall planning process remains unchanged. Therefore, re-clustering is not required in such scenarios.
>
> If APIs are dynamically added, this process can be treated as adding new points to the existing clusters. For dynamic additions, we can adopt an online K-means approach. Specifically, this method uses an exponential moving average of cluster centers to adjust the weights of newly introduced points. This approach ensures that the time complexity of updating for new APIs is relatively low.
>
> If previously available APIs become unavailable, there is no need to modify the original clusters. Instead, an "unavailable" flag can be added when attempting to call the unavailable API, and the next available API can be used as a substitute.
>
> Thank you once again for your feedback on our paper. This will definitely help us improve our work. If these suggestions have helped clarify any doubts you had, we would appreciate it if you could improve the ratings. If there are any unclear points or further details you'd like to discuss, please feel free to respond.

---

> > ### Comment · Reviewer_uydA · 2024-11-26
> >
> > Thank you for your response. I have no other questions, and I am raising my score from 5 to 6.

---

> > > ### Author Response · Authors · 2024-12-02
> > > **Thanks for your warm response**
> > >
> > > Thanks for your support of our work!

---

### Official Review · Reviewer_HRkH · 2024-10-31

**Soundness:** 3
**Presentation:** 2
**Contribution:** 3
**Rating:** 5
**Confidence:** 4

**Summary:**

This paper introduces the Tool-Planner method, a toolkit-based task planning framework designed to enhance the task planning and execution efficiency of LLMs in the context of tool learning. Tool-Planner achieves this by clustering APIs with similar or identical functionalities into toolkits, enabling LLMs to plan across these toolkits. This approach facilitates rapid adjustments and optimization of planning pathways when encountering tool-related errors, thereby improving overall performance.

**Strengths:**

- The proposed method is both intuitive and effective, as it enhances the accuracy and efficiency of LLMs in tool invocation through the clustering of tools and planning at the toolkit level.
- Experimental results on the ToolBench and APIBench datasets substantiate the efficacy of Tool-Planner in improving both success rates and competitive performance.

**Weaknesses:**

- Certain aspects of the methodology require further clarification. For instance, in Section 3.2, the statement "In solving problems for specific states, the model will choose any API within the toolkit for invocation" raises concerns regarding its potential oversimplification. Additionally, since each API necessitates a specific parameter structure for input, how does the model determine the corresponding parameters for an arbitrarily selected API?
- The performance of Tool-Planner is significantly contingent upon the accuracy of the clustering process. Inadequate clustering may hinder the model's ability to identify suitable tools, thereby affecting task completion. Careful tuning of the clustering parameter is essential to achieve optimal performance, which may necessitate additional experimentation and adjustments.
- In the section on "Planning Exploration on Solution Space," it is important to quantify the number of tasks that progress to the "Task Replanning Across Toolkits" stage. Furthermore, what are the corresponding token consumption and computational time for these tasks?
- The ablation study requires further elaboration on the actual configuration of the baseline with Toolkit Integration. Specifically, how is the integration achieved, and what criteria are used to select the toolkit before invoking a specific API? Additionally, an adaptive adjustment method for the number of clusters should be described, along with experimental results validating the use of clustering methods such as DBSCAN for adaptive clustering.
- In the "Efficiency Evaluation" section, it is noted that the runtime of Tool-Planner is significantly higher than that of other algorithms outside of DFSDT. I would like to see a comprehensive analysis of the number of LLM invocations and token usage statistics associated with different methods. This would provide a more thorough assessment of method efficiency. Furthermore, analyzing the impact of cluster quantity selection on execution efficiency would be a valuable addition to the discussion.

**Questions:**

Please refer to the Weaknesses section for details. I’m open to discussion and increasing my score if my comments are addressed satisfactorily.

---

> ### Author Response · Authors · 2024-11-23
> **Response to Reviewer HRkH (1/3)**
>
> We sincerely thank the reviewers for their thorough evaluation of our paper and for providing numerous valuable suggestions across various sections. We also greatly appreciate the positive feedback, such as recognizing that the proposed method is both intuitive and effective, and that the empirical results validate the method's effectiveness. I am deeply grateful for the reviewers' acknowledgment and will now provide detailed explanations for the questions you raised.
>
> > __W1__. Certain aspects of the methodology require further clarification. For instance, in Section 3.2, the statement "In solving problems for specific states, the model will choose any API within the toolkit for invocation" raises concerns regarding its potential oversimplification. Additionally, since each API necessitates a specific parameter structure for input, how does the model determine the corresponding parameters for an arbitrarily selected API?
>
> Thank you very much for raising this point. In fact, we have tried various reordering methods at this stage, but they did not result in significant performance improvements. Through our experiments, we observed that prioritizing APIs within the same toolkit has minimal impact on the performance of large language models in tool-learning scenarios.
>
> Our preliminary analysis suggests that this is because tools with similar functions are grouped within the same toolkit as much as possible. Moreover, tools within a single toolkit are often interchangeable for solving specific types of tasks. For instance, multiple tools can address questions like "What is the weather in New York tomorrow?" or "What was the opening price of the S&P 500 today?" Therefore, the key focus in this process is on selecting and utilizing effective APIs to complete the reasoning along a predetermined planning path. In many cases, reordering the task execution sequence is unnecessary (e.g., approaches like Tool-Planner based on depth-first search). The original planning framework is already sufficient to solve the corresponding problems.
>
> Our approach, which clusters tools, significantly improves and optimizes the time and planning overhead involved in this process. Specifically, after multiple path modifications, models may tend to generate longer plans. For example, while DFSDT might require calling 10 tools to complete a task after several path adjustments, Tool-Planner can solve the same task in just 3-4 steps.
>
> Regarding API parameter selection, the model determines the parameters during the process of extracting and attempting the current API from the toolkit. As elaborated in Section 3.3, during the API invocation and reasoning process, after the model identifies the relevant API and retrieves its documentation, we use the in-context learning (ICL) capability of the LLM to generate the corresponding parameters. Once these parameters are generated, they are passed to the tool for execution, and the result is obtained.
>
> This process is clearly described in the original text. In terms of implementation, when an API is selected, ChatGPT’s function-calling feature generates a structured function call after retrieving the corresponding documentation. This is also clearly demonstrated in our implementation.
>
> > __W2__. The performance of Tool-Planner is significantly contingent upon the accuracy of the clustering process. Inadequate clustering may hinder the model's ability to identify suitable tools, thereby affecting task completion. Careful tuning of the clustering parameter is essential to achieve optimal performance, which may necessitate additional experimentation and adjustments.
>
> Yes, I completely agree with this point. As discussed in our ablation study section on __Impact of different numbers of clusters__, when the number of clusters is too large, the method tends to degrade, reducing its reasoning efficiency. This results in a longer overall reasoning chain, which prevents the generated outcomes from achieving the desired level of quality.
>
> On the other hand, if the number of clusters is too small, clustering fails to group similar methods effectively, leading to an overly generalized reasoning process that cannot focus on the specific sub-tasks that need to be addressed. Therefore, achieving an appropriate clustering balance can significantly improve the efficiency of the reasoning process, reduce the overall length of the reasoning chain, and facilitate better task planning.

---

> ### Author Response · Authors · 2024-11-23
> **Response to Reviewer HRkH (2/3)**
>
> >__W3__. In the section on "Planning Exploration on Solution Space," it is important to quantify the number of tasks that progress to the "Task Replanning Across Toolkits" stage. Furthermore, what are the corresponding token consumption and computational time for these tasks?
>
> I believe your point is very valid, and it highlights one of the key advantages of our approach. Compared to DFSDT, our method significantly reduces the number of replanning instances during the Task Replanning Across Toolkits phase. Since the task replanning process inherently relies on contextual information, the token consumption is inevitably high.
>
> To help you better understand this process, we have conducted a more detailed quantification of the task replanning across toolkits. We evaluated the average API execution time per call, token consumption (for generating configurations) per call, time consumption for planning transitions, and token usage during the planning process transitions. The corresponding token consumption and computation time are as follows(while performance optimal):
>
> |                                  | G1     | G2     | G3     | TorchHub | HuggingFace |
> |----------------------------------|--------|--------|--------|----------|-------------|
> | Avg. API execution Token Cost    | 243.78 | 282.51 | 260.98 | 158.43   | 131.22      |
> | Avg. API execution Time Cost(ms) | 2610   | 2446   | 2371   | 4251     | 3348        |
> | Avg. re-planning token Cost      | 553.22 | 520.91 | 597.12 | 498.33   | 503.41      |
> | Avg. re-planning Time Cost(ms)   | 4210   | 3899   | 4732   | 3871     | 3688        |
> | re-planning proportion           | 6.6%   | 7.3%   | 8.4%   | 4.2%     | 7%          |
>
> As shown, it is crucial to optimize this step effectively. Large language models often tend to rely heavily on previous content during task planning, appending new information at the end or splitting intermediate steps for processing. This behavior leads to an unnecessarily lengthy reasoning process and increases the likelihood of intermediate information errors. However, Tool-Planner effectively mitigates these issues, offering significant advantages in terms of time and efficiency.  These benefits are particularly evident in reducing replanning time and mitigating the propagation of errors caused by replanning.
>
> Furthermore, this clustering-based approach shows great potential when integrated with strategy-learning or planning methods such as ReAct and Reflexion, offering notable improvements in performance.
>
> >__W4__. The ablation study requires further elaboration on the actual configuration of the baseline with Toolkit Integration. Specifically, how is the integration achieved, and what criteria are used to select the toolkit before invoking a specific API? Additionally, an adaptive adjustment method for the number of clusters should be described, along with experimental results validating the use of clustering methods such as DBSCAN for adaptive clustering.
>
> Thank you very much for your suggestions here. Regarding how to implement integration, our approach with ReACT and AdaPlanner replaces the original reasoning components that utilized tools with the following steps: first, we conduct a preprocessing step involving tool clustering; then, we allocate tools from the clustered toolkits to assist the model in completing sub-tasks of the planning process. This type of method is highly adaptable to scenarios that rely on policy-based learning and planning approaches. Consequently, in our ablation experiments, we directly replaced the reasoning components for sub-tasks.
>
> We will more emphasize this section and clarify that the configurations involving the toolkits are designed to replace and compare with the original tool reasoning components. The adaptive adjustment of the number of clusters allows us to fine-tune the clustering granularity within a specific range, using methods such as DBSCAN. Our primary objective is to ensure that the overall reasoning process remains controllable, and predefining the number of clusters helps manage the time required for re-exploring the solution space.

---

> ### Author Response · Authors · 2024-11-23
> **Response to Reviewer HRkH (3/3)**
>
> > __Follow W4__
>
> The advantage of DBSCAN lies in its ability to more accurately group tools with similar functionalities. Considering that the features after encoding contain more unsupervised semantic information in the vector directions, we use cosine similarity as the distance metric for DBSCAN. Default minpts is set to 1. To make a thorough comparison, we evaluated the clustering performance of DBSCAN.
>
> Win Rate:
>
> |                            | $$\epsilon$$ | G1-Inst  | G2-Inst  | G3-Inst  | TorchHub | HuggingFace | TensorFlow |
> |----------------------------|----------|----------|----------|----------|----------|-------------|------------|
> | Tool-Planner(k-means best) | -        | 63.3     | **72.3** | 67.5     | 77.6     | 75.4        | **72.4**   |
> | Tool-Planner(DBSCAN)       | 0.001    | 57.5     | 68.3     | 66.3     | 75.1     | 72.8        | 65.2       |
> | Tool-Planner(DBSCAN)       | 0.01     | 61.8     | 71.8     | 66.5     | 77.3     | 76.5        | 70.3       |
> | Tool-Planner(DBSCAN)       | 0.02     | **64.3** | 70.3     | **68.0** | **78.5** | **77.2**    | 71.9       |
> | Tool-Planner(DBSCAN)       | 0.04     | 63.5     | 69.8     | 68.3     | 77.0     | 73.1        | 69.7       |
> | Tool-Planner(DBSCAN)       | 0.08     | 59.5     | 59.3     | 66.0     | 73.2     | 65.8        | 61.2       |
> | Tool-Planner(DBSCAN)       | 0.16     | 50.5     | 45.5     | 57.5     | 65.1     | 56.9        | 55.9       |
>
> It was observed that DBSCAN achieved slight performance improvements over methods like k-means++. Therefore, in scenarios where performance is the primary focus, DBSCAN can perform better when the number of clusters is appropriate. Conversely, in scenarios that emphasize overall efficiency and consistency, the k-means++ method ensures stable performance and time consumption throughout the process.
>
> > __W5__. In the "Efficiency Evaluation" section, it is noted that the runtime of Tool-Planner is significantly higher than that of other algorithms outside of DFSDT. I would like to see a comprehensive analysis of the number of LLM invocations and token usage statistics associated with different methods. This would provide a more thorough assessment of method efficiency. Furthermore, analyzing the impact of cluster quantity selection on execution efficiency would be a valuable addition to the discussion.
>
>
> Yes, this is an important factor. We provided detailed analysis of the calls and performance comparisons in earlier discussions. Large language models often tend to heavily rely on prior content during task planning, appending new information at the end or splitting intermediate steps for processing. This behavior can lead to unnecessarily lengthy reasoning processes and increase the likelihood of errors in intermediate information.
>
> Tool-Planner effectively mitigates these issues, offering significant advantages in terms of time and efficiency. These benefits are particularly evident in reducing re-planning time and alleviating error propagation caused by re-planning. Additionally, we have supplemented the analysis with the specific impact of DBSCAN clustering on execution efficiency, providing a more comprehensive empirical and analytical evaluation of the reasoning process and solution efficiency.
>
>
> Thank you once again for your feedback on our paper. This will definitely help us improve our work. If these suggestions have helped clarify any doubts you had, we would appreciate it if you could improve the ratings. If there are any unclear points or further details you'd like to discuss, please feel free to respond.

---

> ### Author Response · Authors · 2024-12-02
> **Potential Follow up to Reviewer HRkH**
>
> Dear Reviewer,
>
> Thank you for your detailed review and the many constructive questions you've raised. We have provided detailed explanations to address your concerns. As the period of Author Response deadline is approaching, we are expecting to receive your feedback. Your suggestions will greatly contribute to the improvement of our work and guide the future direction of our research. If our responses have addressed your concerns, we kindly hope for your consideration in improving the score. Once again, we sincerely appreciate your valuable assistance and support to our work!

---

### Official Review · Reviewer_TwKU · 2024-11-02

**Soundness:** 3
**Presentation:** 3
**Contribution:** 3
**Rating:** 8
**Confidence:** 4

**Summary:**

This paper introduces a new framework, Tool-Planner, which formulates LLM planning with API tools as a tree-search over clusters of tools, or 'toolkits', generated using SimCSE. Experiments and ablations show the soundness of this approach over its closest relative, DFSDT, which does a tree-search over single tools.

**Strengths:**

$Originality:$ This paper builds upon DFSDT and introduces a toolkit which provides better performance.

$Quality$: The experiments cover domains with many APIs, relevant baselines, and ablations which showcase the importance of the design choices.

$Clarity$: The paper is presented very clearly, especially methodology. My only nitpick is Figure 2 - my first  impression was that this was another Toolchain [1] paper because the search seems to be over single APIs rather than a selection over toolkits and then the most relevant API in the toolkit.

$Significance$: LLM abilities are empowered as they gain access to new tools but they also perform poorly in reasoning tasks with large action spaces. This paper provides a clever trick to cluster tools together, reducing the LLM decision space and simplifying the planning task. This will ensure that LLM performance in domains with tools remains relatively high regardless of the future introduction of many new tools.

[1] ToolChain*: Efficient Action Space Navigation in Large Language Models with A* Search (Zhuang et al. 2023)

**Weaknesses:**

- Table 1's feature categories are vague (what is Tool Integration?) and don't make it obvious how your work  clearly contrasts from DFSDT. "Tool Clustering" distinguishes this work more from DFSDT.
- Figure 2 looks like DFSDT or ToolChain from a glance; it is not obvious that the nodes were chosen from toolkits. Adding more emphasis / text on the toolkits could make things clearer.
- Toolchain is compared against in Table 1 but missing from the experiments

**Questions:**

- How were the "Explainable Planning" and "Tool Integration" categorizations decided?
- What does "Tool Integration" refer to?
- How often do you observe failures where prioritizing APIs within the same toolkit leads to unnecessary expansions (due to the appropriate API being in another toolkit)?
- Why is ToolChain compared against in Table 1 but not in the experiments?

---

> ### Author Response · Authors · 2024-11-23
> **Response to Reviewer TwKU (1/3)**
>
> Thank you very much to the reviewer for the series of praises for our work and the extremely high evaluation, including recognizing the importance of our design choices, commending the clarity of the paper's presentation, and appreciating the clever technique provided in this work to integrate tools. Your insightful and detailed comments have greatly helped us refine and enhance our work. Moving forward, I will focus on addressing the concerns and questions you have raised.
>
> > __Clarity__. My only nitpick is Figure 2 - my first impression was that this was another Toolchain [1] paper because the search seems to be over single APIs rather than a selection over toolkits and then the most relevant API in the toolkit.
>
> Our figure aims to illustrate how the paper achieves the following during the reasoning process: 1) selecting a toolkit and executing the corresponding reasoning capability, and 2) quickly updating the selected tools within the toolkit to fix errors while maintaining the same logical process. The search process treats the toolkit as a whole. For instance, among the available toolkits presented by the recipe access APIs, if two tools within the current toolkit are unavailable, another toolkit's API can be selected to accomplish the task of retrieving recipe information.
>
> Toolchain* approach adopts a heuristic search scheme based on the A* algorithm. This method requires reasoning over the solution space and evaluating the next steps at every stage, which introduces significant delays in execution efficiency. This approach differs significantly from ours.
>
> > __W1__. Table 1's feature categories are vague (what is Tool Integration?) and don't make it obvious how your work clearly.
>
> Tool integration refers to the process of aggregating multiple tools with similar functionalities to achieve the same results, rather than separating them. The key distinction in our work lies in the use of a tool-clustering approach, which groups tools with the same functionality under an initial planning framework for reasoning. This reasoning framework identifies intermediate cluster points as corresponding processing nodes. For each cluster point, we select the method that currently guarantees successful and effective results as the intermediate output within the planning framework. To the best of our knowledge, we are the first to adopt such a tool-clustering approach, and its effectiveness has been demonstrated across multiple levels.
>
> > __W2__. Figure 2 looks like DFSDT or ToolChain from a glance; it is not obvious that the nodes were chosen from toolkits. Adding more emphasis / text on the toolkits could make things clearer.
>
> Thank you very much for your suggestions. In the current version of our diagram, we use thumbnails to illustrate the correspondence between toolkits and the tools within them, while the tree structure represents the effectiveness and reasoning process in the planning workflow. We will further optimize the structure of the diagram and incorporate emphasis, labels, and annotations to more clearly highlight that the nodes are selected from the toolkits.
>
> > __W3__. Toolchain is compared against in Table 1 but missing from the experiments
>
> Thank you very much for bringing this up. Approaches like ToolChain* and other heuristic-based tool-learning methods, such as LATS, leverage heuristic search (e.g., MCTS) or reinforcement learning strategies to achieve better global planning in complex multi-step reasoning tasks. However, these methods face two core challenges. First, their heuristic algorithms require evaluating potential solutions globally (each new solution requires invoking the model for reasoning). Second, these methods fail to address how to handle alternative paths effectively when the corresponding API encounters an issue. These methods simply attempt to reason through all possible paths (using heuristics for search) and assign a very low score when encountering an error. While this approach avoids errors when selecting a path during a given step, it does not address the problem of avoiding repeated issues with faulty APIs during subsequent reasoning attempts. As a result, the same API error from a previous attempt is likely to reoccur during the next reasoning process.
>
> In contrast, strategy learning or planning-based methods, such as AdaPlanner, DFSDT, and ToolPlanner, can process tasks more efficiently and respond to errors with rule-based corrections, minimizing additional iterations caused by errors. These methods also provide a more interpretable decision-making process during planning. Considering our goal of achieving faster and more convenient task planning while obtaining optimal reasoning results within limited inference time, strategy learning or planning-based approaches align better with our requirements.

---

> ### Author Response · Authors · 2024-11-23
> **Response to Reviewer TwKU (2/3)**
>
> > __Follow W3__
>
> As we mentioned earlier, approaches like ToolChain* rely on algorithms like A* as their foundation, requiring extensive tool invocation analysis during path exploration. In the experiments described in the original paper, ToolChain* operates in scenarios with a very small set of categories, where only a limited number of tools (10–100, as stated in section D.3 of the ToolChain* paper) are available. In such constrained settings, the ToolChain* method can be effective. However, when the number of tools increases, such as in real-world scenarios or broader datasets like APIBench or ToolBench that encompass multiple tasks, ToolChain*'s performance significantly deteriorates or even becomes ineffective.
>
> Therefore, we focused primarily on a detailed performance comparison of strategy learning or planning-based approaches in the previous section. Nevertheless, to ensure a fair comparison, we also compared ToolChain*'s performance in two aspects: 1) within the specific subtask of "Home Search" and 2) across the entire dataset. The comparative results are shown in the table.
>
>
> Win Rate:
> | Model  | Method       | Home Search | G1-Inst | G1-Tool | G1-Cat | G2-Inst | G2-Cat | G3-Inst |
> |-|-|-|-|-|-|-|-|-|
> | GPT3.5 | DFSDT        | 71.0  | 58.3    | 59.8    | 56.8   | 70.5    | 59.3   | 68.0    |
> | GPT3.5 | ToolChain*   | 69.0  | 56.3    | 58.5    | 53.0   | 67.5    | 58.0   | 61.5    |
> | GPT3.5 | **Tool-Planner** | 75.0  | 63.3    | 63.5    | 61.3   | 72.3    | 63.5   | 67.5    |
> | GPT4   | DFSDT        | 73.0  | 65.8    | 69.3    | 65.3   | 72.0    | 56.8   | 81.5    |
> | GPT4   | ToolChain*   | 76.0  | 38.5    | 70.5    | 64.0   | 73.3    | 54.0   | 75.5    |
> | GPT4   | **Tool-Planner** | 79.0  | 75.5    | 75.8    | 71.8   | 79.8    | 70.3   | 92.0    |
>
> Pass Rate:
>
> | Model  | Method       | Home Search | G1-Inst | G1-Tool | G1-Cat | G2-Inst | G2-Cat | G3-Inst |
> |-|-|-|-|-|-|-|-|-|
> | GPT3.5 | DFSDT        | 86.0        | 48.5    | 62.5    | 58.0   | 71.5    | 70.5   | 61.0    |
> | GPT3.5 | ToolChain*   | 90.0        | 22.5    | 28.0    | 27.0   | 32.5    | 28.0   | 17.0    |
> | GPT3.5 | **Tool-Planner** | 90.0        | 58.5    | 71.0    | 66.0   | 75.5    | 78.0   | 66.0    |
> | GPT4   | DFSDT        | 87.0        | 57.0    | 72.0    | 64.5   | 77.5    | 69.5   | 71.5    |
> | GPT4   | ToolChain*   | 91.0        | 31.8    | 33.5    | 27.5   | 41.0    | 31.8   | 19.5    |
> | GPT4   | **Tool-Planner** | 92.0        | 66.0    | 78.5    | 75.0   | 83.5    | 77.5   | 83.0    |
>
> As seen from the results, our method consistently outperforms the ToolChain* algorithm in both performance and effectiveness. This demonstrates the robustness and efficacy of our approach.
>
> > __Q1__. How were the "Explainable Planning" categorizations decided?
>
> Interpretable planning primarily evaluates whether the reasoning process of our method is interpretable. Similar to approaches such as LATS and ToolChain*, their planning processes are primarily based on strategies like evaluation functions or reinforcement learning, which require reasoning across multiple paths simultaneously to derive the next layer of reasoning results. Moreover, methods like ToolChain*, which adopt BFS-based A* algorithms, extend the reasoning process to explore other possible shorter paths. In contrast, methods like ReAct and DFSDT, which include reasoning explanations at every step and execute the reasoning and decision-making processes sequentially, follow a DFS-like reasoning approach. This makes them highly interpretable and more easily adjustable. This interpretability is largely based on the ability to trace back from the reasoning results to understand the decision-making process at each step.
>
> > __Q2__. How were the "Tool Integration" categorizations decided? What does "Tool Integration" refer to?
>
> Tool integration primarily refers to whether the framework classifies a given set of tools. Our approach integrates and combines tools based on tool clustering, whereas other methods generally focus on analyzing and handling tools from the perspective of a single tool. To a significant extent, our approach outperforms other methods in both reasoning performance and time efficiency.

---

> ### Author Response · Authors · 2024-11-23
> **Response to Reviewer TwKU (3/3)**
>
> > __Q3__. How often do you observe failures where prioritizing APIs within the same toolkit leads to unnecessary expansions (due to the appropriate API being in another toolkit)?
>
> In fact, our experiments did observe this phenomenon: within the same toolkit, the priority ordering of APIs has little impact on the results of large language models in tool-learning scenarios. Our preliminary analysis suggests this is because we group APIs with similar functionalities into the same toolkit whenever possible. When the clustering is stable, additional reordering has a limited impact on performance and results. If the initial setup includes an excessive number of clusters, some APIs might indeed end up in a different toolkit, but this is not the main factor affecting performance. The primary issue with multiple clusters is a significant decline in efficiency. Therefore, clustering APIs with similar functionalities greatly enhances both performance and efficiency.
>
> > __Q4__. Why is ToolChain* compared against in Table 1 but not in the experiments?
>
> Kindly see the response in W3. when the number of tools increases, such as in real-world scenarios or broader datasets like APIBench or ToolBench that encompass multiple tasks, ToolChain*'s performance significantly deteriorates or even becomes ineffective.
>
> Thank you once again for your feedback on our paper. This will definitely help us improve our work. If there are any unclear points or further details you'd like to discuss, please feel free to respond.

---

> > ### Comment · Reviewer_TwKU · 2024-11-26
> >
> > Thank you for engaging with my concerns, I am satisfied with the responses. I would still appreciate if the authors can improve Figure 2 clearer in the paper. I will be maintaining my score.

---

> > > ### Author Response · Authors · 2024-12-02
> > > **Thanks for your warm response**
> > >
> > > Thanks for your support of our work, we would incorporate the updated version for the Figure 2 to better illustrate the two main concepts: 1) selecting a toolkit and executing the corresponding reasoning capability, and 2) quickly updating the selected tools within the toolkit to fix errors while maintaining the same logical process. Thanks for your warm response.

---

### Official Review · Reviewer_L3vz · 2024-11-08

**Soundness:** 3
**Presentation:** 3
**Contribution:** 2
**Rating:** 5
**Confidence:** 4

**Summary:**

This paper proposes clustering similar tool APIs via LLMs and embeddings to enhance LLM planning efficiency in tool use. The ToolPlanner framework effectively addresses tool planning and invocation tasks using toolkits. Empirical results demonstrate the framework's superiority in performance and efficiency compared to multiple baselines, while ablations confirm the effectiveness of tool clustering.

**Strengths:**

1. The paper is well-written and easy to follow.
2. Comprehensive ablation studies on the core component, the toolkit.
3. Detailed appendix with experimental specifics and case studies.
4. Insightful error analysis.

**Weaknesses:**

1. The paper could include more comparisons with tree-based inference methods in LLMs, e.g., [1][2] in related work.
2. Consider comparing with ToolChain*[3].

**Questions:**

1. I suggest a name change for the framework to better reflect its toolkit-based planning, e.g., ToolkitPlanner.
2. Beyond Figure 4, what is the computation cost/wall time for clustering these APIs?
3. Line 217: How is another available API t′ selected within the same toolkit?
4. Line 259: Which LLM is used to assess Win Rate?
5. Line 262: Could the authors provide more details on detecting and measuring hallucinations?
6. The novelty is somewhat limited, as noted in Line 375, where the method is derived from DFSDT by replacing API nodes with toolkits. The authors could elaborate more in Sec 3.3 and include task replanning experiments to better distinguish this framework from DFSDT. While on the other hand, extensive results and ablations strengthen confidence in the framework's efficiency and effectiveness

 [1] Zhou, Andy, et al. "Language Agent Tree Search Unifies Reasoning, Acting, and Planning in Language Models." Forty-first International Conference on Machine Learning.

 [2] Feng, Xidong, et al. "Alphazero-like tree-search can guide large language model decoding and training." arXiv preprint arXiv:2309.17179 (2023).

 [3] Zhuang, Yuchen, et al. "ToolChain*: Efficient Action Space Navigation in Large Language Models with A* Search." The Twelfth International Conference on Learning Representations.

---

> ### Author Response · Authors · 2024-11-22
> **Response to Reviewer L3vz (1/2)**
>
> Thank you very much to the reviewers for their detailed reading and review of our paper, and for recognizing multiple aspects of our work, including the well-written and easy-to-understand manuscript, comprehensive ablation studies, detailed appendices, and insightful error analysis. I deeply appreciate the acknowledgment of these contributions. I will now address the weaknesses you highlighted in detail:
>
> > __W1__. The paper could include more comparisons with tree-based inference methods in LLMs, e.g., [1][2] in related work.
>
> Thank you for suggesting these two papers. [1] employs an AlphaZero-like tree search approach, combined with learned value functions, to guide the decoding and training of large language models (LLMs). [2] uses Monte Carlo Tree Search (MCTS), where the tree search identifies potentially high-value paths, and MCTS selects optimal actions at each step based on value assessments. These methods leverage heuristic search techniques like MCTS or reinforcement learning strategies, enabling better global planning in complex multi-step reasoning tasks. However, a key limitation of these approaches lies in the need to evaluate possible global solutions using heuristic algorithms, where every new solution requires invoking the model for inference. Additionally, these methods do not adequately address the issue of handling failed API calls. When encountering errors, these approaches typically assign a very low score to the erroneous path, which, while preventing repeated errors in the current path selection, does not resolve issues with the API call when the error recurs. As a result, the model often has to reattempt problematic API calls in subsequent inferences. In contrast, methods based on strategy learning or planning, such as AdaPlanner, DFSDT, and ToolPlanner, are more efficient and can handle tasks quickly. They also incorporate rules to correct errors and minimize unnecessary error iterations, while the planning process itself offers better interpretability. Considering our objective is to complete task planning efficiently and achieve more accurate reasoning results within a limited inference time, strategy learning or planning-based approaches are more suitable for our needs.
>
>
> > __W2__. Consider comparing with ToolChain*[3].
>
> Thank you for this suggestion. As mentioned earlier, ToolChain*-like methods rely on A* algorithms and require analysis through multiple tool invocations during path exploration. The experiments in their original paper are conducted in relatively small categories, with a limited number of tools (only 10–100, as noted in section D.3 of the ToolChain* paper). Therefore, ToolChain* can perform effectively in these scenarios. However, when the number of tools increases, as in real-world scenarios or comprehensive benchmarks like APIBench or ToolBench, the performance of ToolChain* significantly degrades and may even become ineffective. Thus, our primary focus has been on comparing with strategy learning or planning-based methods in terms of detailed performance. Nevertheless, for a fair comparison, we have included ToolChain*'s performance on 1) the Home Search subtask and 2) the overall dataset. The comparison results are shown in the table:
>
> Win Rate:
> | Model  | Method       | Home Search | G1-Inst | G1-Tool | G1-Cat | G2-Inst | G2-Cat | G3-Inst |
> |-|-|-|-|-|-|-|-|-|
> | GPT3.5 | DFSDT        | 71.0  | 58.3    | 59.8    | 56.8   | 70.5    | 59.3   | 68.0    |
> | GPT3.5 | ToolChain*   | 69.0  | 56.3    | 58.5    | 53.0   | 67.5    | 58.0   | 61.5    |
> | GPT3.5 | **Tool-Planner** | 75.0  | 63.3    | 63.5    | 61.3   | 72.3    | 63.5   | 67.5    |
> | GPT4   | DFSDT        | 73.0  | 65.8    | 69.3    | 65.3   | 72.0    | 56.8   | 81.5    |
> | GPT4   | ToolChain*   | 76.0  | 38.5    | 70.5    | 64.0   | 73.3    | 54.0   | 75.5    |
> | GPT4   | **Tool-Planner** | 79.0  | 75.5    | 75.8    | 71.8   | 79.8    | 70.3   | 92.0    |
>
> Pass Rate
>
> | Model  | Method       | Home Search | G1-Inst | G1-Tool | G1-Cat | G2-Inst | G2-Cat | G3-Inst |
> |-|-|-|-|-|-|-|-|-|
> | GPT3.5 | DFSDT        | 86.0        | 48.5    | 62.5    | 58.0   | 71.5    | 70.5   | 61.0    |
> | GPT3.5 | ToolChain*   | 90.0        | 22.5    | 28.0    | 27.0   | 32.5    | 28.0   | 17.0    |
> | GPT3.5 | **Tool-Planner** | 90.0        | 58.5    | 71.0    | 66.0   | 75.5    | 78.0   | 66.0    |
> | GPT4   | DFSDT        | 87.0        | 57.0    | 72.0    | 64.5   | 77.5    | 69.5   | 71.5    |
> | GPT4   | ToolChain*   | 91.0        | 31.8    | 33.5    | 27.5   | 41.0    | 31.8   | 19.5    |
> | GPT4   | **Tool-Planner** | 92.0        | 66.0    | 78.5    | 75.0   | 83.5    | 77.5   | 83.0    |
>
> It is shown that as the number of tools increases, the pass rate of heuristic-based methods significantly declines due to the massive reasoning required, while tool-planner maintains competitive performance both on win/pass rate in the final results.

---

> ### Author Response · Authors · 2024-11-22
> **Response to Reviewer L3vz (2/2)**
>
> > __Q1__. I suggest a name change for the framework to better reflect its toolkit-based planning, e.g., ToolkitPlanner.
>
> Thank you for the insightful suggestion regarding the name of our framework. We agree that a name emphasizing the toolkit-based planning aspect could better convey the essence of our approach. However, our current name was chosen to highlight not only the planning mechanism but also our method's emphasis on efficient and adaptive decision-making processes. Also, the planning process remains fundamentally driven by the reasoning and planning processes involving tools. Nevertheless, we acknowledge the potential benefits of a more intuitive name, such as ToolkitPlanner, which clearly reflects the framework's nature. We will carefully consider this recommendation and evaluate whether a name change could enhance the clarity and impact of our framework's presentation. If so, we will ensure that the revised name effectively represents the key features of toolkit-based planning while aligning with our overall research objectives.
>
> > __Q2__. Beyond Figure 4, what is the computation cost/wall time for clustering these APIs?
>
> Our framework employs the kmeans++ clustering method, which benefits from the use of a uniform initialization strategy. Given the substantial number of relevant points included in our tool types, the computational cost of clustering is relatively favorable. In our calculations, under the theoretically most time-consuming scenario (with 16,464 tools and 128 clusters), the clustering inference achieves an average runtime of 8.2 seconds on an 8-thread CPU. For scenarios with different numbers of clusters, the time efficiency improves further. Compared to the total runtime of downstream inference tasks, this portion of the computational cost is relatively minor. Even under a standard single-thread configuration, the inference for the most complex theoretical case only takes 25 seconds to complete.
>
> > __Q3__. Line 217: How is another available API t′ selected within the same toolkit?
>
> Under the same toolkit, we directly utilize another tool that has not been used during the current toolkit invocation. For example, $ t_0 $ to $  t_{i-1} $  represent the tools we have already used, and $t_i$ is the tool we are currently using. If $  t_i $  becomes unavailable or encounters an error, we randomly select and use one from $  t_{i+1} $  to $  t_n $ .
>
> > __Q4__. Line 259: Which LLM is used to assess Win Rate?
>
> gpt-4-turbo-2024-04-09 is used to evaluate the performance of the two answers. This setup is chosen to ensure a fair comparison. It aligns with the configuration used in the ToolLLM method, specifically the DFSDT approach.
>
> > __Q5__. Line 262: Could the authors provide more details on detecting and measuring hallucinations?
>
> RapidAPI provides a corresponding AST-based evaluation framework. In this framework, hallucination is defined as the case where the generated function call corresponds to a tool that is entirely imaginary. Additionally, during evaluation, we classify any newly introduced parameters—parameters that do not exist in the actual function—as hallucinations. Based on these two criteria, we evaluate the process of function call generation and its final results across different methods.
>
> > __Q6__. The novelty is somewhat limited, as noted in Line 375, where the method is derived from DFSDT by replacing API nodes with toolkits. The authors could elaborate more in Sec 3.3 and include task replanning experiments to better distinguish this framework from DFSDT.
>
> We sincerely appreciate the reviewer’s suggestions regarding this section of our paper. While our iterative process for exploring potential solutions does share similarities with DFSDT-derived methods, the novelty of our work extends significantly beyond that scope. To the best of our knowledge, we are the first to employ clustering to integrate and classify tool information. By grouping tools based on their specific functionalities, we create collections that assist large language models in offering rapid and diverse options for reasoning and planning in specific scenarios or tasks.
>
> In Section 3.3, we provide a detailed explanation of how the introduced clustering approach informs the process of tool selection. We will further clarify the method for switching tools within the same toolset and provide formalized explanations on how to switch between clustered toolkits and re-plan tasks more effectively.
>
> In Appendix E, we include various scenarios that might arise, accompanied by examples of task re-planning. These examples, along with the experimental results, strongly demonstrate how grouping multiple available tools and relying on pre-existing reasoning paths enhance the tool learning process for reasoning tasks.
>
> Thank you once again for your feedback on our paper. If these suggestions have helped clarify any doubts you had, we would appreciate it if you could improve the ratings. please feel free to respond.

---

> > ### Author Response · Authors · 2024-12-02
> > **Potential Follow up to Reviewer L3vz**
> >
> > Dear Reviewer,
> >
> > Thank you for your detailed review and the many constructive questions you've raised. We have provided detailed explanations to address your concerns. As the period of Author Response deadline is approaching, we are expecting to receive your feedback. Your suggestions will greatly contribute to the improvement of our work and guide the future direction of our research. If our responses have addressed your concerns, we kindly hope for your consideration in improving the score. Once again, we sincerely appreciate your valuable assistance and support to our work!

---

### Author Response · Authors · 2024-12-02
**General Response for Readers(2/2)**

|                                  | G1     | G2     | G3     | TorchHub | HuggingFace |
|----------------------------------|--------|--------|--------|----------|-------------|
| Avg. API execution Token Cost    | 243.78 | 282.51 | 260.98 | 158.43   | 131.22      |
| Avg. API execution Time Cost(ms) | 2610   | 2446   | 2371   | 4251     | 3348        |
| Avg. re-planning token Cost      | 553.22 | 520.91 | 597.12 | 498.33   | 503.41      |
| Avg. re-planning Time Cost(ms)   | 4210   | 3899   | 4732   | 3871     | 3688        |
| re-planning proportion           | 6.6%   | 7.3%   | 8.4%   | 4.2%     | 7%          |


> 3. Impact of Other Clustering Algorithms (e.g., DBSCAN)

DBSCAN remains robust and highly effective within our framework. Its advantage lies in accurately grouping tools with similar functions. Compared to k-means++, DBSCAN performs better under stricter constraints, maintaining strong performance. However, overly stringent constraints may reduce the number of tools meeting these criteria, potentially degrading performance to a level similar to DFSDT. Therefore, in performance-critical scenarios, DBSCAN achieves better results when the number of clusters is appropriate. Conversely, in scenarios emphasizing overall efficiency and consistency, k-means++ ensures stable performance and time efficiency throughout. The corresponding results are shown below:

Win Rate:

|                            | $$\epsilon$$ | G1-Inst  | G2-Inst  | G3-Inst  | TorchHub | HuggingFace | TensorFlow |
|----------------------------|----------|----------|----------|----------|----------|-------------|------------|
| Tool-Planner(k-means best) | -        | 63.3     | **72.3** | 67.5     | 77.6     | 75.4        | **72.4**   |
| Tool-Planner(DBSCAN)       | 0.001    | 57.5     | 68.3     | 66.3     | 75.1     | 72.8        | 65.2       |
| Tool-Planner(DBSCAN)       | 0.01     | 61.8     | 71.8     | 66.5     | 77.3     | 76.5        | 70.3       |
| Tool-Planner(DBSCAN)       | 0.02     | **64.3** | 70.3     | **68.0** | **78.5** | **77.2**    | 71.9       |
| Tool-Planner(DBSCAN)       | 0.04     | 63.5     | 69.8     | 68.3     | 77.0     | 73.1        | 69.7       |
| Tool-Planner(DBSCAN)       | 0.08     | 59.5     | 59.3     | 66.0     | 73.2     | 65.8        | 61.2       |
| Tool-Planner(DBSCAN)       | 0.16     | 50.5     | 45.5     | 57.5     | 65.1     | 56.9        | 55.9       |

Finally, we deeply appreciate the reviewers' efforts and the recognition of our work. If there are further questions or suggestions regarding our methodology, please feel free to response.

---

### Author Response · Authors · 2024-12-02
**General Response for Readers(1/2)**

Thank you all for your thorough review and valuable suggestions on our paper. We believe that these recommendations will not only contribute to improving the paper in the future but also have a profound impact on the adoption of the proposed method. Here, we aim to re-emphasize the advantages and unique aspects of our work while comprehensively addressing the common concerns raised by reviewers and readers:

> 1. Further Advancement in Tool Learning

Our method represents a significant step forward in the field of tool learning. Rather than simply providing tools to LLMs and allowing them to choose, we propose selecting the most representative toolkit from a broader range of options and utilizing it for problem-solving and planning. This approach enables efficient identification of usable tools during planning and maintains stability throughout the reasoning process. It minimizes error propagation in long reasoning chains, thereby ensuring higher output quality. This method extends the strategy-learning paradigm in planning while retaining efficient inference capabilities.

> 2. Clustering as a Planning Strategy in Tool Learning

Our clustering-based approach aligns well with realistic planning processes and addresses the inefficiencies seen in existing methods, such as DFSDT and AnyTool (representing strategy-learning approaches), and heuristic methods like ToolChain*. Specifically, it reduces the high latency caused by frequent calls to multiple tools in large action spaces. For scenarios involving thousands of tools, our method achieves an optimal balance between efficiency and execution time. This has been validated on the two most commonly used datasets for tool learning with the largest number of tools available. Furthermore, the clustering mechanism ensures scalability, limiting additional overhead even as the number of tools grows.

__Common Concerns from Reviewers__:

> 1. Performance Comparison with Heuristic Planning Methods (e.g., ToolChain*)

ToolChain* relies on the A* algorithm and performs multiple tool calls during path exploration. Experiments in the original ToolChain* paper were conducted in relatively small categories with a limited number of tools (only 10-100, as detailed in Section D.3 of their paper). While ToolChain* performs well in such scenarios, its performance degrades significantly, even becoming ineffective, when the number of tools increases—such as in real-world applications or comprehensive benchmarks like APIBench and ToolBench. Corresponding experimental results are provided below:

Win Rate:
| Model  | Method       | Home Search | G1-Inst | G1-Tool | G1-Cat | G2-Inst | G2-Cat | G3-Inst |
|-|-|-|-|-|-|-|-|-|
| GPT3.5 | DFSDT        | 71.0  | 58.3    | 59.8    | 56.8   | 70.5    | 59.3   | 68.0    |
| GPT3.5 | ToolChain*   | 69.0  | 56.3    | 58.5    | 53.0   | 67.5    | 58.0   | 61.5    |
| GPT3.5 | **Tool-Planner** | 75.0  | 63.3    | 63.5    | 61.3   | 72.3    | 63.5   | 67.5    |
| GPT4   | DFSDT        | 73.0  | 65.8    | 69.3    | 65.3   | 72.0    | 56.8   | 81.5    |
| GPT4   | ToolChain*   | 76.0  | 38.5    | 70.5    | 64.0   | 73.3    | 54.0   | 75.5    |
| GPT4   | **Tool-Planner** | 79.0  | 75.5    | 75.8    | 71.8   | 79.8    | 70.3   | 92.0    |

Pass Rate

| Model  | Method       | Home Search | G1-Inst | G1-Tool | G1-Cat | G2-Inst | G2-Cat | G3-Inst |
|-|-|-|-|-|-|-|-|-|
| GPT3.5 | DFSDT        | 86.0        | 48.5    | 62.5    | 58.0   | 71.5    | 70.5   | 61.0    |
| GPT3.5 | ToolChain*   | 90.0        | 22.5    | 28.0    | 27.0   | 32.5    | 28.0   | 17.0    |
| GPT3.5 | **Tool-Planner** | 90.0        | 58.5    | 71.0    | 66.0   | 75.5    | 78.0   | 66.0    |
| GPT4   | DFSDT        | 87.0        | 57.0    | 72.0    | 64.5   | 77.5    | 69.5   | 71.5    |
| GPT4   | ToolChain*   | 91.0        | 31.8    | 33.5    | 27.5   | 41.0    | 31.8   | 19.5    |
| GPT4   | **Tool-Planner** | 92.0        | 66.0    | 78.5    | 75.0   | 83.5    | 77.5   | 83.0    |

> 2. Cost of Tool-Planner in Terms of Computation

This is a key advantage of our approach. Compared to DFSDT, our method significantly reduces the number of re-planning instances across toolkits. Since re-planning inherently relies on context, token consumption during this phase is inevitably high, making optimization essential. Large language models often exhibit a strong dependency on prior content during task planning, adding new information at the end or splitting intermediate steps. This leads to unnecessarily lengthy reasoning processes and increases the likelihood of errors in intermediate information. However, Tool-Planner effectively mitigates these issues, providing significant advantages in time and efficiency. These benefits are especially notable in reducing re-planning time and minimizing error propagation caused by re-planning. Relevant experiments are as follows:

---

### Meta-Review · Area_Chair_rNFT · 2024-12-22

**Metareview:**

The submission introduces Tool(kit?)-Planner, an approach that helps LLMs with planning chains of tool use by first clustering similar tools into "toolkits" and then running search in the space of toolkits. The paper provides a useful error analysis (section 4.4)  and an empirical evaluation that shows the approach provides significant gains in task completion rates compared to search over the original space of tools.

Strengths:

- Simple and intuitive idea, easy to implement
- Insightful empirical evaluation
- Potential for impact
- Clarity of the submission

Weaknesses:

- The dependence of performance on tool clustering quality, including the choice of the number of clusters

The metareviewer recommends this work for publication due to the practicality and effectiveness of its approach. Tool-Planner's dependence on clustering quality is indeed a conceptual issue, but in practice, after the clustering is tuned once from scratch for a given system, the clusters can be subsequently updated incrementally as new tools become available, which is generally a less error-prone process.

**Additional Comments On Reviewer Discussion:**

The discussion focused largely on the dependence on clustering quality, certain aspects of empirical analysis, and addition questions about Tool-Planner's behavior. The authors' responses mostly succeeded in addressing these issues. (In the case of two reviewers, who didn't respond to the authors' rebuttals, the metareviewer had to make the call whether their concerns were addressed.)

---

### Decision · Program_Chairs · 2025-01-22

Accept (Poster)